# Turbofan Engine Health Assessment Based on Spatial–Temporal Similarity Calculation

**DOI:** 10.3390/s23249748

**Published:** 2023-12-11

**Authors:** Cheng Peng, Xin Hu, Zhaohui Tang

**Affiliations:** 1School of Computer, Hunan University of Technology, Zhuzhou 412007, China; pengcheng@hut.edu.cn; 2School of Automation, Central South University, Changsha 410083, China; zhtang128@163.com

**Keywords:** turbofan engine, sequence matching, similarity calculation, remaining useful life

## Abstract

Aiming at the problem of the remaining useful life prediction accuracy being too low due to the complex operating conditions of the aviation turbofan engine data set and the original noise of the sensor, a residual useful life prediction method based on spatial–temporal similarity calculation is proposed. The first stage is adaptive sequence matching, which uses the constructed spatial–temporal trajectory sequence to match the sequence to find the optimal matching sample and calculate the similarity between the two spatial–temporal trajectory sequences. In the second stage, the weights of each part are assigned by the two weight allocation algorithms of the weight training module, and then the final similarity is calculated by the similarity calculation formula of the life prediction module, and the final predicted remaining useful life is determined according to the size of the similarity and the corresponding remaining life. Compared with a single model, the proposed method emphasizes the consistency of the test set and the training set, increases the similarity between samples by sequence matching with other spatial–temporal trajectories, and further calculates the final similarity and predicts the remaining use through the weight allocation module and the life prediction module. The experimental results show that compared with other methods, the root mean square error (RMSE) index and the remaining useful life health score (Score) index are reduced by 12.6% and 14.8%, respectively, on the FD004 dataset, and the RMSE index is similar to that in other datasets; the Score index is reduced by about 10%, which improves the prediction accuracy of the remaining useful life and can provide favorable support for the operation and maintenance decision of turbofan engines.

## 1. Introduction

With the continuous improvement of industrialization, the structure of industrial equipment has become increasingly complex. For the aviation industry, the aircraft engine, as the most important core component of an aircraft, directly determines whether the aircraft can operate stably and reliably. Due to the harsh environment of high temperature and high pressure that the engine is subjected to for long periods, it is prone to engine performance degradation. At the same time, as a highly sophisticated thermomechanical device, repairing a failed engine is extremely difficult, leading to irreparable losses. The turbofan engine, as a typical type of aircraft engine, is used in various types of aircraft. However, due to its complex operating conditions and large amount of data, predicting the remaining life of a turbofan engine is quite challenging. Therefore, how to extract effective features from the data and accurately predict the remaining useful life of an engine has become a hot topic and a difficult problem in the field of industrial life prediction.

There are three types of existing remaining useful life (RUL) prediction models: model-based methods, data-driven methods, and hybrid methods [1,2]. Model-based methods include physical model-based methods and statistical model-based methods. The physical model-based methods for RUL prediction are mainly derived from reliability theory research. They analyze a large amount of experimental data obtained throughout the lifecycle of mechanical equipment, and then utilize mathematical statistics and probability theory to analyze and process the data, thereby obtaining statistical predictions of RUL based on reliability criteria. Kim et al. [3] proposed a physics-based Markov chain model to identify degradation pathways of lithium-ion batteries. By analyzing the direct correlations between phenomena and states, they used capacity measurements to identify degradation pathways and predict the remaining useful life. Statistical model-based methods, also known as empirical model-based methods, estimate the RUL of machinery by establishing statistical models based on empirical knowledge. Usually, the RUL prediction results are presented as conditional probability density functions based on observation results. Zhang et al. [4] proposed a statistical feature fusion method based on statistical quantities for equipment health condition assessment. This method has advantages in terms of decoupling indicators and fusing multi-source information. The aforementioned model-based methods not only require measured parameters from actual engineering systems but also rely on extensive prior knowledge [5] during the model construction process. They depend on certain levels of expert experience, which may hinder their applicability in transfer learning and domain development.

With the continuous development of artificial intelligence, machine learning [6] and deep learning [7] are gradually being applied to life prediction. At the same time, due to the widespread use of sensors, data-driven [8] methods are receiving more and more attention as it is easier to obtain monitoring data for devices. Ren et al. [9] proposed an adaptive sensor weighting (TGE-ASW) method based on a time-varying Gaussian encoder to address the issue of difficulty in building representative features for multiple sensor raw signals with noise. They used an adaptive sensor weighting strategy and built a convolutional neural network (CNN) to predict RUL by obtaining advanced feature representations. Chen et al. [10] addressed the problem of little research using graph neural networks (GNN) to capture spatial correlations between sensors, by introducing sensor embedding and proposing a new RUL prediction model based on ConvGAT. Hu et al. [11] proposed a deep bidirectional recursive neural network (DBRNNs) integration method. In this method, several DBRNNs with different neuron structures were constructed to extract hidden features from sensory data for predicting the remaining useful life of aircraft engines. Li et al. [12] addressed the problem of increasing complexity of degradation characteristics of aircraft engine components during flight with multi-operating work points (MOP) and proposed a deep learning fusion algorithm based on a self-attention mechanism (SAM). This algorithm uses a one-dimensional CNN to extract spatial features and a long short-term memory(LSTM) network [13,14] to fuse the measurement data of one component, and extracts time features from actual measured data.

Combining model-based and data-driven methods [15] into hybrid approaches can overcome the limitations of both methods. However, due to the complexity and high cost of the models, the development of hybrid methods has not been very successful, and there is limited research in this area. Hybrid algorithms aim to address the shortcomings of the model-based approaches by combining them with data-driven methods. Khumprom et al. [16] used an evolutionary selection method to choose features from the C-MAPSS aircraft gas turbine engine dataset, and then applied the selected features to train a hybrid convolutional long short-term memory (CNN-LSTM) deep neural network for RUL prediction. XueBin et al. [17] proposed a diagnostic method for predicting remaining useful life based on degradation trajectory similarity. They first accurately constructed degradation trajectories using convolutional neural network autoencoders and attention mechanisms. Then, they used a new similarity matching rule to evaluate the similarity of degradation trajectories. The results showed that this method has good predictive performance and low sensitivity to sample size, and can be easily incorporated into similarity-based frameworks.

Although neural networks have shown promising performance in predictive tasks, there are several challenges in current research:(1)The impact of spatial characteristics on mechanical life prediction may vary in different working environments. In such cases, it is a challenging problem to integrate time features and spatial features to improve trajectory similarity.(2)Due to the presence of noise and other interferences in the raw signals from sensors, there may be issues with improper allocation of feature weights during similarity calculation, thereby affecting the final prediction results.

To address the aforementioned issues, this study proposes a life prediction model based on spatial–temporal similarity calculation, aiming to enhance the accuracy of RUL predictions. Firstly, certain features exhibit minimal variation throughout the entire time span and carry little information. Including all features directly in the model would result in longer training time. Therefore, this study adopts an adaptive feature selection method to eliminate features that remain unchanged during the lifecycle, thus resolving the issue of data redundancy. Secondly, within the selected features, spatial characteristics are identified, and a modified longest common subsequence (LCSS) algorithm is utilized to calculate the similarity of spatial–temporal trajectories, thereby improving the accuracy of similarity calculation. Finally, the weight training module of the life prediction model is used to assign the feature weights of the remaining parts, and then the final prediction RUL is generated by the life prediction module of the life prediction model.

## 2. Definition of Terms

The dataset used in this study is the NASA dataset, where each set of degradation trajectory data consists of the engine ID, rounds, three configuration parameters, and measurement data from 21 sensors. Prior to adaptive matching, certain preprocessing steps need to be applied to the dataset [18]. Due to the uniqueness of the dataset and the subsequent algorithm descriptions, it is necessary to define relevant terms required for the algorithms.

**Definition** **1.**
*Spatial–Temporal Trajectory Sequence.*


The environmental type of an engine can be represented by a triad of attributes, namely, flight altitude, Mach number, and flight speed, defined as envor = {param_1, param_2, param_3}, where param_1 represents configuration parameter 1, param_2 represents configuration parameter 2, and param_3 represents configuration parameter 3. The set of environmental types is defined as E = {envor_1, envor_2, …, envor_n}, where n is the total number of environmental types. A spatial–temporal event at time t is represented as a tuple event = (t, e), where t is the occurrence time of the event and e is an environmental type from the set of environmental types E. A spatial–temporal trajectory sequence is represented as L = {event_1, event_2, …, event_n}, where n is the total number of space-time events in the trajectory sequence.

**Definition** **2.**
*Remaining Useful Life (RUL) Metric.*


Since the dataset does not provide a specific indicator for lifespan, after analyzing the dataset, the number of rounds from the engine’s healthy state to failure state is considered as the lifespan indicator. The formula for calculating the RUL is defined as follows: (1)$$RUL=\max(T_{i})-1+t$$
where *i* represents the current engine ID, *T* represents the operating sequence of the current engine ID, $\max(T_{i})$ represents the maximum number of flight cycles for engine with ID, and *t* represents the flight cycles at the current time.

**Definition** **3.**
*Matching Result Set.*


During the matching process of the matching algorithm, in order to better explain the matching process, the spatial–temporal trajectory sequence of the test set engines is defined as the original string (initial), and the spatial–temporal trajectory sequence of the training set sample engines is defined as the mother string (haystack). After each matching operation, there can be either a successful match or a failed match. When a successful match occurs, a substring (needle) is obtained. The matching cycles are defined as fitCycle = {c_1, c_2, …, c_n}, where n is the length of the substring, and c_1, c_2, c_n correspond to each cycle of the substring. The matching sequence is defined as fitSeries = {number, fitCycle_1, fitCycle_2, …, fitCycle_n}, where number represents the engine ID of a successful match, and the subsequent n matching cycles are the matching cycles of n successful matches. The matching results are defined as fitResult = {fitSeries_1, fitSeries_2, …, fitSeries_n}, where n is the number of successful matches. The matching result set is defined as fitresultSet = {fitResult_1, fitResult_2, …, fitResult_n}, where n is the number of engines in the test set. The matching result set contains the matching results of all engines in the test set and will be used for subsequent similarity calculation algorithms.

**Definition** **4.**
*Spatial–Temporal Similarity.*


To calculate the similarity of sequence matching, the following formula is defined: (2)$$Sim_{LCSS}=\frac{LCSS(needle,haystack)\times 2}{len_{needle}+len_{haystack}}\times \frac{len_{needle}}{len_{initial}}$$
where $Sim_{LCSS}$ is the space–time similarity; neddle is the substring when a successful match occurs; haystack is the mother string when a successful match occurs; initial is the initial substring; LCSS(needle,haystack) is the length of the longest common subsequence between needle and haystack; $len_{needle}$, $len_{haystack}$, $len_{initial}$ are the lengths of needle, haystack, initial, respectively.

**Definition** **5.**
*Sensor Parameter Error.*


In order to ensure that the parameters of each sensor are not affected by dimensionality, data normalization is performed. The following formula is used to restrict the data to the range [0, 1.0]: (3)$$x_{scale}=\frac{x-u}{S}$$
where *x* is the value to be normalized, $x_{scale}$ is the normalized value. *u* represents the mean of the sample, and *S* represents the standard deviation of the sample.

To calculate the sensor parameter error for a successful match, using the matching sequence obtained from the engine ID and matching cycles to calculate the Euclidean distance between the matching sequence and the matching sequence in the training set samples. The Euclidean distance for each parameter of these two sequences is then calculated based on the weights obtained from PCA weighting. This calculation yields the sensor parameter error for a successful match.
(4)$$\delta =\frac{\sum_{i=1}^{n}\sqrt{\sum_{j=1}^{8}W_{j}{\left(nparm_{j}-hparm_{j}\right)}^{2}}}{n}$$
where *i* represents one of the *n* engine numbers, $W_{j}$ represents the weight coefficient for sensor parameter *j*, and $nparm_{j}$ and $hparm_{j}$, respectively, represent the sensor parameters of the substring needle and the mother string haystack with index *j*.

**Definition** **6.**
*Similarity Calculation Formula.*


After performing the above operations and obtaining the spatial–temporal similarity and sensor parameter error, firstly the sensor parameter error is converted into sensor parameter similarity. To combine the similarities of different parts [19,20], a weighted training model is used to allocate weights to these two similarities. In order to compare the similarity of each matching cycle, a similarity formula is defined.
(5)$$Sim=\frac{W_{s}Sim_{LCSS}+W_{\delta }e^{-\delta }}{\max(Sim_{LCSS},e^{-\delta })}$$
where $Sim_{LCSS}$ is the spatial–temporal similarity; $e^{-\delta }$ is the sensor parameter similarity; $W_{s}$, $W_{\delta }$ are weights assigned to $Sim_{LCSS}$ and $e^{-\delta }$, respectively; $\max(Sim_{LCSS}$, $e^{-\delta })$ represents the maximum value between $Sim_{LCSS}$ and $e^{-\delta }$.

## 3. Methods of This Study

### 3.1. Overall Flowchart

The flowchart of the life prediction model based on similarity calculation is shown in Figure 1. The specific steps are as follows:

Step 1.To preprocess the training set and test set data, feature selection is performed using clustering methods and PCA dimensionality reduction. This process also retains the PCA weights of the selected sensor parameters.Step 2.After feature selection, the data are standardized to ensure consistent scales. Meanwhile, spatial–temporal sequences are generated from the preprocessed data for use in subsequent matching algorithms.Step 3.The matching result set for all training set engines is obtained using the improved LCSS algorithm.Step 4.The matching result set obtained in Step 3 is utilized to calculate the spatial–temporal similarity and sensor parameter similarity for each successful match.Step 5.The training module of CRITIC weights is iterated using the spatial–temporal similarity and sensor parameter similarity obtained in Step 4. After completing the training, the weights for each component are determined.Step 6.The final similarity is calculated and the corresponding RUL using the life prediction module is obtained.Step 7.The test set for testing is selected and the evaluation metrics are calculated.

### 3.2. Similarity Calculation Algorithm

After applying the LCSS matching algorithm, we can obtain a set of successful matching sequences from the result set. Within each successful matching sequence, the first element represents the engine ID in the training dataset when the match is successful. The subsequent elements in the matching sequence correspond to all the rounds in the training dataset that match with the target sequence. By analyzing this set, we can extract important information such as the length of the sequence and the rounds in which the sequence occurs in the training dataset. Once we have obtained this information, we can use the similarity calculation Algorithm 1 defined in this study to calculate spatial–temporal similarity and sensor parameter errors. The algorithm inputs include the matching result (fitResult), individual engine data from the test set units (testEngineData), and the all training set units (trainSet). The fitResult records critical matching information, while testEngineData can obtain the initial length of the sequence and calculate sensor parameter errors based on the matching result and the sensor parameters of the trainSet. The calculated spatial–temporal similarity and sensor parameter errors are then used as inputs for the weight training model. The weight training model uses these inputs to train the weights. After got the training weights, we can obtain the final similarity descending table and corresponding predicted RUL by applying the weights to calculate the similarity.
**Algorithm 1** Similarity Calculation Algorithm**Require:** Training set trainSet, Test engine data testEngineData, Matching result fitResult**Ensure:** Sorted similarity and corresponding predicted RUL {Sim, RUL}    m←fitResult.length    **for** int i = 1 to m **do**        fitSeries←fitResult[i]        turbine_number←fitSeries[0]        n←fitSeries.length        **for** j = 1 to n **do**            fitCycle←fitSeries[j]            lenneedle←fitCycle.length            lenhystack←fitCyle[last]−fitCycle[first]+1            leninitial←testEngineData.length            $Sim_{LCSS}=\frac{LCSS(needle,haystack)\;\times \;2}{len_{needle}\;+\;len_{haystack}}\times \frac{len_{needle}}{len_{initial}}$            $\delta =\frac{\sum_{i=1}^{n}\sqrt{\sum_{j=1}^{8}W_{j}{\left(nparm_{j}\;-\;hparm_{j}\right)}^{2}}}{n}$            $Sim=\frac{W_{s}Sim_{LCSS}\;+\;W_{\delta }e^{-\delta }}{\max(Sim_{LCSS},e^{-\delta })}$        **end for**    **end for**    return $\left\{\begin{array}{c} Sim,RUL \end{array}\right\}$


The calculation process is as follows:(a)Retrieve the length of the match result set, fitResult.(b)Iterate through the match result set to extract the matching sequences. Obtain the engine ID in the training dataset when a match is successful, as well as the length of the matching sequence.(c)Calculate the lengths of the substring (needle) and the main string (haystack) by taking the difference between the length of the match rounds and the starting point. Obtain the initial substring length, initial, from the test engine data in the input. Use the formula to calculate the spatial–temporal similarity and sensor parameter errors between the sequences.(d)Train the weights for each component using the calculated spatial–temporal similarity and sensor parameter errors. Use these weights in the lifespan prediction module to obtain the final sequence similarity.(e)Return the sequence similarity descending table.

## 4. Experiment and Result Analysis

### 4.1. Introduction to Experimental Platforms and Datasets

The experimental environment for this study is shown in Table 1.

In order to verify the predictive performance of the proposed model, using NASA’s prognastics centre of excellence (PCoE) turbofan condition monitoring data commercial modular aeropropulsion system simulation software (C-MAPSS) [21], we simulated the whole process from operation to failure of turbofan engines under different environmental conditions to obtain experimental data sets. The structural model of the turbofan engine is shown in Figure 2. The dataset includes simulated data of the full life cycle of the turbofan engine and the remaining life value collected at a certain moment. The dataset consists of four sets of data: FD001, FD002, FD003, and FD004, each of which is collected under different operating conditions and fault modes, as shown in Table 2.

Each set of degradation trajectory data consists of an engine ID, rounds, three types of setting parameters, and measurements from 21 sensors, as described in Table 3. In the experiment, the cycle number is used to reconstruct the RUL of the turbofan engine. The impact of the 3 setting parameters and 21 sensor measurements on the RUL can exhibit positive correlation, negative correlation, no correlation, or uncertain relationship. Therefore, it is necessary to perform feature selection on the data [22] to eliminate irrelevant variables, identify important features, and reduce computational burden [23]. Based on the above definition, the degradation trajectory data of the first engine in the FD004 dataset was analyzed, and it was determined that the three setting parameters are correlated with the RUL. Combined with a correlation analysis of the sensor parameters in Figure 3, the final selection included sensors 7, 8, 9, 12, 13, 14, 20, and 21 as the usable features.

### 4.2. Analysis of Experimental Process and Results

After performing data preprocessing such as feature selection and standardization on the dataset, spatial–temporal sequences are generated and training samples are constructed. The engine data in the test set are then matched with the training samples based on sequence matching; the matching process is shown in Figure 4.

Matching process:(a)Obtain the spatial–temporal trajectory sequences of the test engine and all the spatial–temporal trajectory sequences of the training samples.(b)Check if the spatial–temporal trajectory sequence of the test engine exceeds the maximum length of the spatial–temporal trajectory sequences in the training samples. If it does not exceed the maximum length, the spatial–temporal sequence can be directly generated. If it exceeds the maximum length, it needs to be truncated before generating the spatial–temporal sequence.(c)Perform sequence matching between the current spatial–temporal trajectory sequence and the spatial–temporal trajectory sequences in the training samples. If the match is successful, the process ends. Otherwise, truncate the first round of the current sequence and perform sequence matching again until a successful match is found.(d)After a successful match, record the engine number and matching round for each successful match. Package all the matching rounds to obtain the matching result set, which will be used for subsequent similarity calculation and weight training models.

Before applying the sequence matching algorithm, it is necessary to generate the spatial–temporal trajectory sequences. The specific parameters for the environmental types in various spatial–temporal events are shown in Table 4:

After generating the spatial–temporal sequences that meet the criteria, we can proceed with the sequence matching against the spatial–temporal sequences in the training set. Let us assume that the test sequence is ABBCD and the training sequence is AFBBDB. The matching process is shown in Table 5 below. Eventually, we can obtain the longest common subsequence of these two sequences as ABBD.

After completing the sequence matching, the spatial–temporal similarity and sensor parameter errors are calculated using a formula. These values are then used as inputs for weight training module. Once the weight training is completed, the remaining useful life prediction can be obtained using the life prediction module. The comparison between the predicted and real values for the FD001, FD002, FD003, and FD004 datasets is shown in Figure 5, Figure 6, Figure 7 and Figure 8.

From the above comparison between predicted and real values, it can be seen that the predicted RUL is quite close to the real RUL labels. This validates the effectiveness and feasibility of the proposed method. To further evaluate the prediction performance of the model, this paper adopts two evaluation metrics to assess the reasonability and accuracy of the predicted results. For an engine degradation scenario an early prediction is preferred over late predictions. RMSE reflects the degree of fit between the predicted RUL and the real RUL, with a smaller score indicating better performance. And Score reflects the reasonableness of the RUL predictions, also with a smaller score indicating better performance. Different from RMSE, Score pays more attention to the true time of failure, and uses different parameters in the function to control the magnitude of penalties for late and early predictions. It can be observed that there is a more severe penalty for late predictions. It is expressed as follows: (6)$$Score=\left\{\begin{array}{cc} \sum_{i=1}^{N}\left(e^{-\frac{E_{i}}{10}}-1\right), & E_{i}<0\\ \sum_{i=1}^{N}\left(e^{-\frac{E_{i}}{13}}-1\right), & E_{i}\geq 0 \end{array}\right.$$
(7)$$RMSE=\sqrt{\frac{1}{N}\sum_{i=1}^{N}E_{i}^{2}}$$
where $E_{i}$ is the difference between the real RUL and the predicted RUL numbered *i*, and *N* is the total number. From the above two formulas, as $E_{i}$ becomes smaller, both the Score and RMSE become smaller, indicating more accurate predictions and better forecasting performance. Therefore, the primary objective of the prediction algorithm is to minimize the prediction error $E_{i}$ as much as possible, aiming for $E_{i}$ to be equal to 0.

### 4.3. Comparative Experimental Analysis

In order to verify the prediction accuracy of the proposed method, the predicted results of the four datasets were compared with other methods such as support vector regression (SVR), convolutional neural networks (CNN), long short-term memory (LSTM), and gradient boosting decision trees (GBDT) using two evaluation metrics, RMSE and Score. The kernel function for SVR is selected as the radial basis kernel function. The CNN uses a sliding window of length 15 and a stride of 1 to construct a two-dimensional matrix, consisting of two convolutional layers, two pooling layers, and one standard feedforward neural network layer. The LSTM consists of two LSTM layers with 32 nodes each, and two standard feedforward neural network layers with 8 nodes each. The number of leaf nodes for the improved GBDT algorithm is set to 31. The comparison chart of evaluation metrics is shown in Figure 9 and Figure 10, and the specific results are shown in Table 6 and Table 7.

Based on the comparison and analysis of the above two metrics, the proposed method achieved good prediction performance on different datasets, showing a significant decrease in the Score metric compared to other existing algorithms. In terms of RMSE, there was a significant drop in the FD004 dataset, while the other datasets showed similar results. Specifically, the RMSE and Score metrics decreased by 12.6% and 14.8%, respectively, on the FD004 dataset, while on the other datasets, the RMSE metric remained largely consistent, and the Score metric decreased by about 10%.

To test the universality of the proposed method, we applied it to the rolling bearing dataset [24], which consists of publicly available accelerated performance degradation experimental data for the full lifespan of four rolling bearings from the University of Cincinnati [25]. To increase the sample size, we randomly selected 100 time points within each bearing’s lifespan, resulting in a training set of 400 samples [26,27]. After making slight modifications to some formulas in the method described in this paper [28,29], we conducted experiments and the results are shown in Table 8, all predicted RUL is within the margin of error of the real RUL [30,31,32]. The above experiments and results demonstrate that the method proposed in this paper can be extended to other industrial processes.

## 5. Conclusions

In order to comprehensively consider the high-dimensional and multitype state data characteristics of complex equipment, such as aviation turbofan engines, this paper utilizes the spatial–temporal features in the dataset. By employing an adaptive matching algorithm, it aims to ensure the consistency of spatial–temporal trajectory sequence similarity. Subsequently, the weights are determined through a weight training module based on the matching sequences obtained from adaptive matching. The final similarity is calculated using the lifespan prediction module, which also determines the RUL prediction value. The proposed algorithm is tested on the NASA turbofan engine dataset and compared with other prediction methods, demonstrating its effectiveness and feasibility.

In future work, it can be considered to incorporate anomaly detection techniques to reduce the length of sequences, thereby saving matching time in the adaptive matching algorithm and significantly reducing time costs. Additionally, alternative methods or models can be explored to determine the coefficients for spatial–temporal similarity and error, which may lead to more accurate prediction results. 

## Figures and Tables

**Figure 1 sensors-23-09748-f001:**
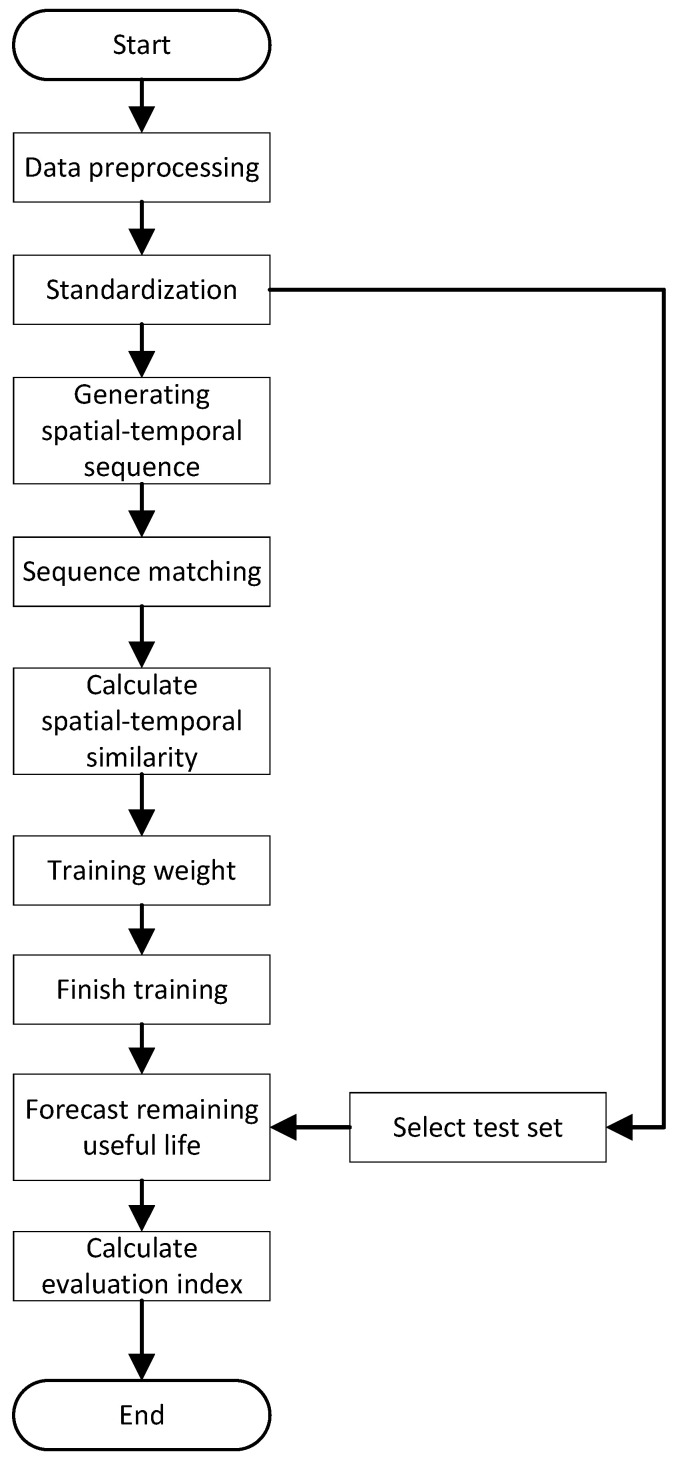
Overall flowchart of the model.

**Figure 2 sensors-23-09748-f002:**
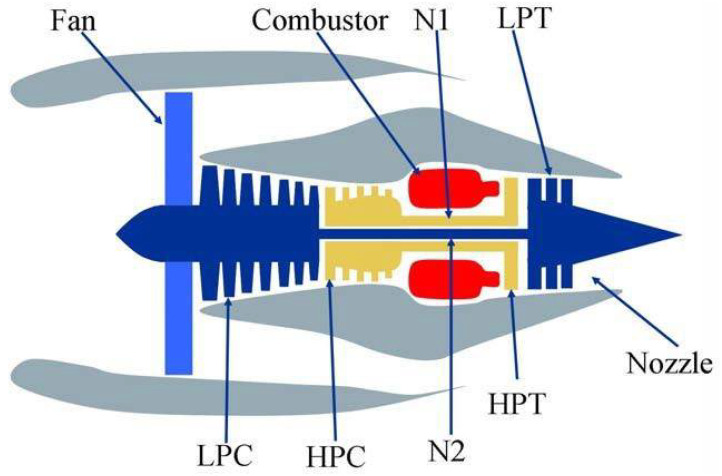
Turbofan engine structure model.

**Figure 3 sensors-23-09748-f003:**
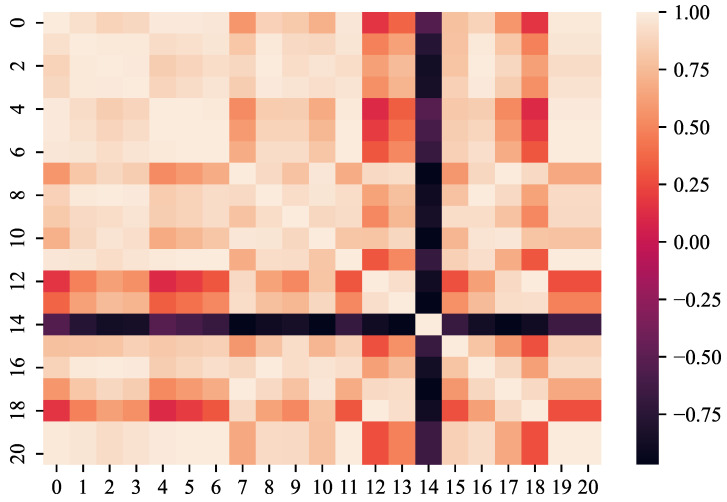
Sensor parameter heat map.

**Figure 4 sensors-23-09748-f004:**
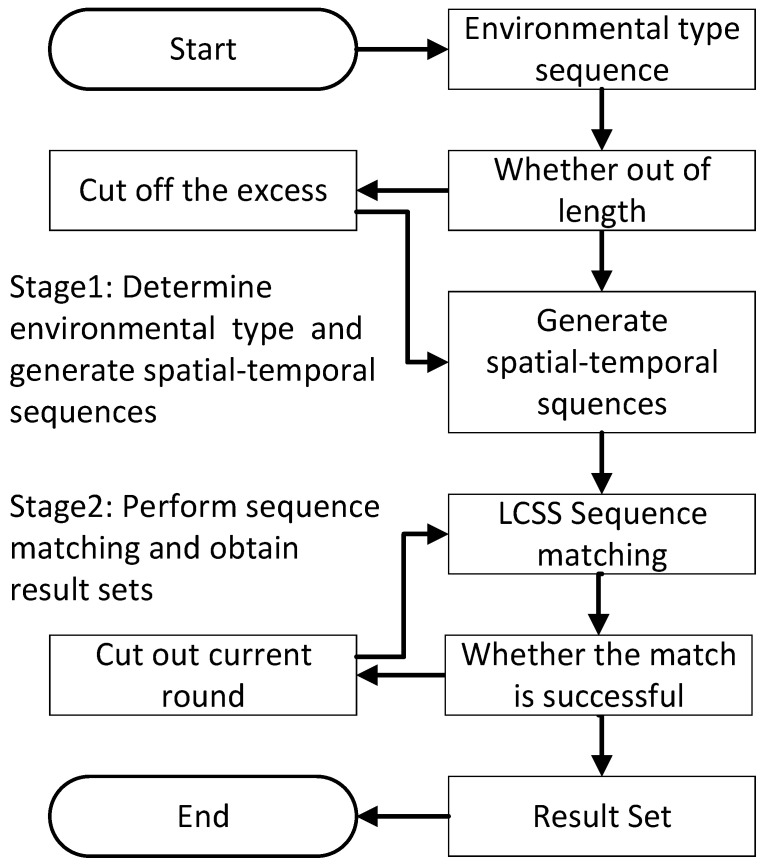
Sequence matching algorithm flowchart.

**Figure 5 sensors-23-09748-f005:**
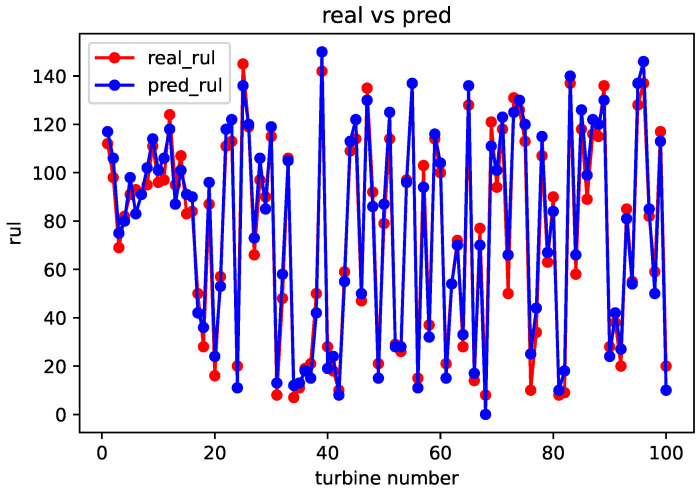
Comparison of predicted value and real value of FD001 dataset.

**Figure 6 sensors-23-09748-f006:**
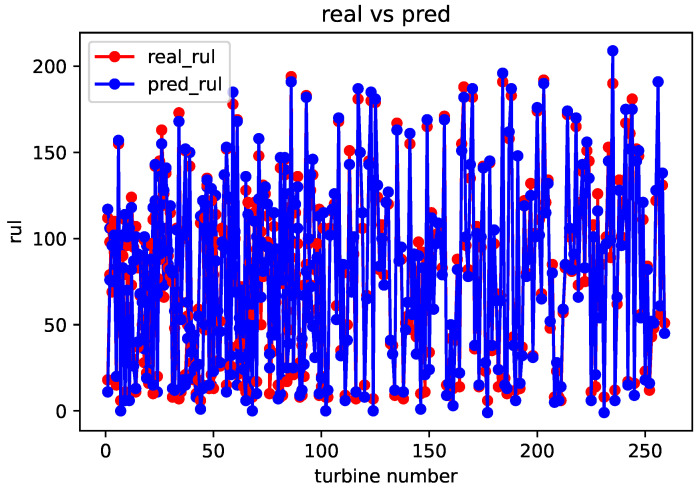
Comparison of predicted value and real value of FD002 dataset.

**Figure 7 sensors-23-09748-f007:**
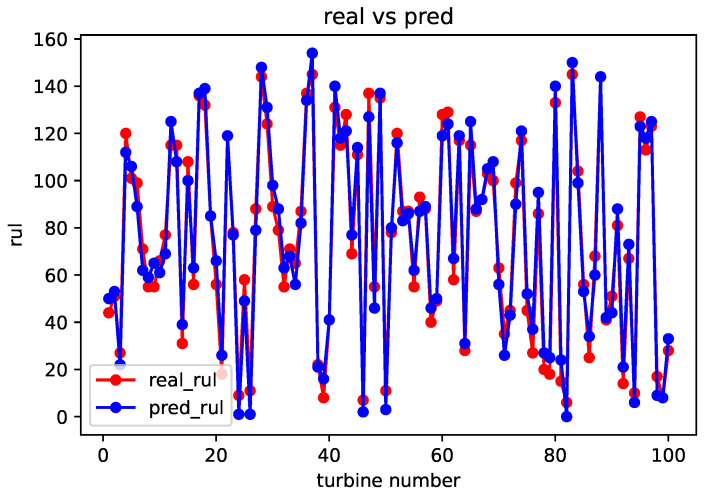
Comparison of predicted value and real value of FD003 dataset.

**Figure 8 sensors-23-09748-f008:**
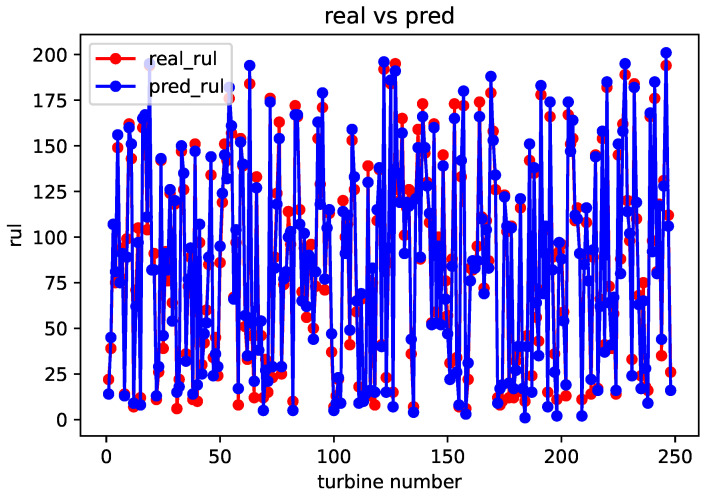
Comparison of predicted value and real value of FD004 dataset.

**Figure 9 sensors-23-09748-f009:**
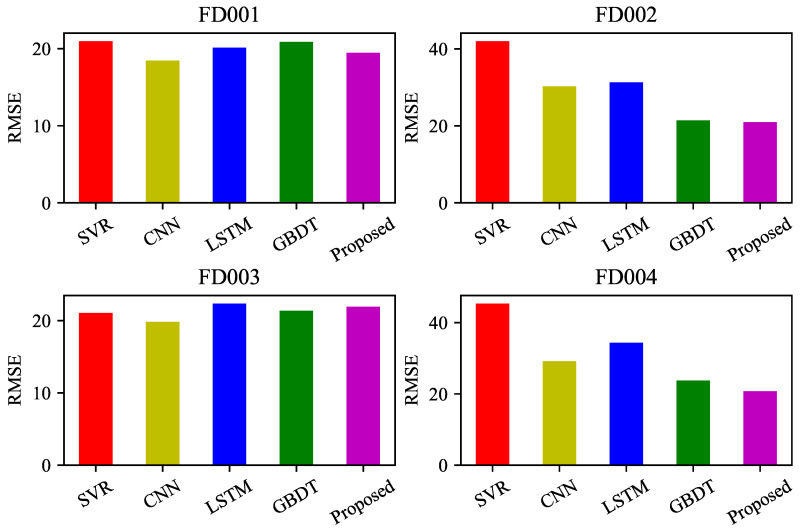
RMSE comparison of different forecasting methods.

**Figure 10 sensors-23-09748-f010:**
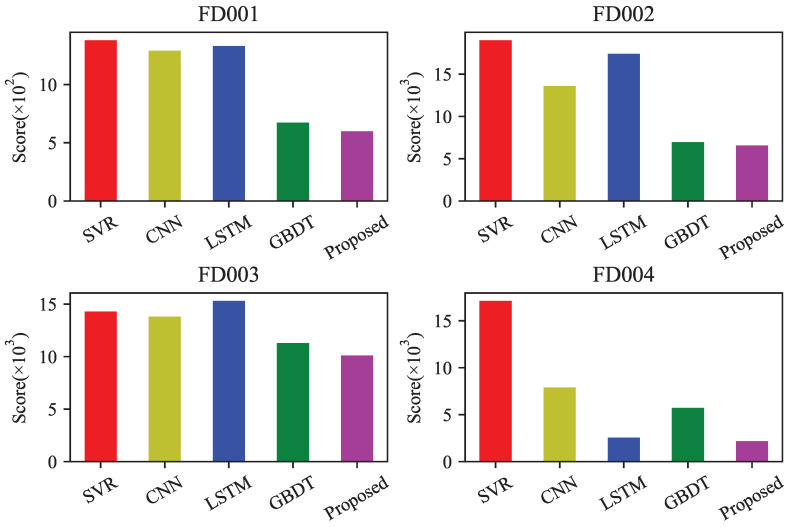
Score comparison of different forecasting methods.

**Table 1 sensors-23-09748-t001:** Experimental environment.

Item	Parameter
Operating system	Windows10
CPU	CoreTMi7—6820HK
Frequency	2.70 Ghz
Storage	240G + 240G
Memory	16G
Algorithm simulation	Python3.9.0, Anaconda4.2.0

**Table 2 sensors-23-09748-t002:** C-MAPSS datase.

DataSets	Train Set Units	Test Set Units	Conditions	Fault Modes
FD001	100	100	1	1
FD002	260	259	6	1
FD003	100	100	1	2
FD004	248	249	6	2

**Table 3 sensors-23-09748-t003:** Description of turbofan engine sensor parameters.

Index	Description	Symbol
1	Total temperature at fan inlet	°C
2	Total temperature at LPC outlet	°C
3	Total temperature at HPC outlet	°C
4	Total temperature at LPT outlet	°C
5	Pressure at fan inlet	psia
6	Total pressure in bypass-duct	psia
7	Total pressure at HPC outlet	psia
8	Physical fan speed	rpm
9	Physical core speed	rpm
10	Engine pressure ratio (P50/P2)	–
11	Static pressure at HPC outlet	psia
12	Ratio of fuel flow to Ps30	pps/psi
13	Corrected fan speed	rpm
14	Corrected core speed	rpm
15	Bypass ratio	–
16	Burner fuel–air ratio	–
17	Bleed enthalpy	–
18	Demanded fan speed	rpm
19	Demanded corrected fan speed	rpm
20	HPT coolant bleed	lbm/s
21	LPT coolant bleed	lbm/s

**Table 4 sensors-23-09748-t004:** Environment type parameter table.

Number	Height	Mach Number	Velocity
A	0±0.01	0±0.01	100
B	10±0.01	0.25±0.01	20
C	20±0.01	0.70±0.01	0
D	25±0.01	0.62±0.01	80
E	35±0.01	0.84±0.01	60
F	42±0.01	0.84±0.01	40

**Table 5 sensors-23-09748-t005:** The matching process for the sequence matching algorithm.

	A	F	B	B	D	B
A	1	1	1	1	1	1
B	1	1	2	2	2	2
B	1	1	2	3	3	3
C	1	1	2	3	3	3
D	1	1	2	3	4	4

**Table 6 sensors-23-09748-t006:** RMSE comparison of different forecasting methods.

Method	FD001	FD002	FD003	FD004
SVR	20.96	42	21.05	45.35
CNN	18.45	30.29	19.82	29.16
LSTM	20.13	31.30	22.37	34.34
GBDT	20.88	21.40	21.37	23.75
Proposed	19.47	20.95	25.94	20.75

**Table 7 sensors-23-09748-t007:** Score comparison of different forecasting methods.

Method	FD001	FD002	FD003	FD004
SVR	$1.38\times 10^{3}$	$1.90\times 10^{4}$	$1.43\times 10^{3}$	$1.71\times 10^{4}$
CNN	$1.29\times 10^{3}$	$1.36\times 10^{4}$	$1.38\times 10^{3}$	$7.89\times 10^{3}$
LSTM	$1.33\times 10^{3}$	$1.74\times 10^{4}$	$1.53\times 10^{3}$	$2.56\times 10^{3}$
GBDT	$6.73\times 10^{2}$	$6.95\times 10^{3}$	$1.13\times 10^{3}$	$5.73\times 10^{3}$
Proposed	$5.99\times 10^{2}$	$6.57\times 10^{3}$	$1.01\times 10^{3}$	$2.18\times 10^{3}$

**Table 8 sensors-23-09748-t008:** The experinment result of rolling bearing.

Number of Bearing	Predicted RUL/h	Real RUL/h	Error/%
Data_3rd_1	29	25	16.0
Data_3rd_2	12	14	14.3
Data_3rd_3	29	28	3.6
Data_3rd_4	24	27	11.1

## Data Availability

The data presented in this study are openly available in NASA Prognostics Data Repository at 10.1109/PHM.2008.4711414, reference number [21].

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
