# Peer review of "Turbofan Engine Health Assessment Based on Spatial–Temporal Similarity Calculation"

_sensors, 2023, doi:10.3390/s23249748_

Round 1

Reviewer 1 Report

Comments and Suggestions for Authors

The paper “Turbofan engine health assessment based on spatial-temporal similarity calculation” considers the spatial-temporal features in the dataset in order to comprehensively analyse the high-dimensional and multi-type state data characteristics of complex technical object. The approach is tested on the NASA dataset for aviation turbofan engine.

In line 14, “the RMSE index and Score index” the abbreviations are not clarified.

An adaptive data analysis algorithm was used based on the method of comparing the similarity of spatiotemporal sequences of trajectories.

Please check the terminology in Fig.3 and Fig.4 (“spatiotemporal”, but in the paper “spatial-temporal”).

Algorithm title (after Line 212) is duplicated (Algorithm 1 Algorithm 1:) 

Variable “trainSet” in Line 2 of the algorithm 1 is not defined in the head of the algorithm.

Variable “data data” in Line 2 of the algorithm 1 is not clear.

As a result, the comparison of trajectories solves the problem of adjusting analysis algorithms to build a learning module. Next, the prediction module is used to solve the similarity analysis problem. The performance of the proposed approach is tested on a NASA turbofan data set and compared with other prediction methods, demonstrating its effectiveness and similarity.

In further research, the authors propose to use the proposed solutions to detect anomalies in a data set and exclude them to reduce the length of the sequences under study.

The content of the Conclusion section reflects the main content of the article.

The Reference list can be expanded to 30-35 links.

Tables and figures are appropriate.

The remarks are also marked in color in the attached paper file. Please see the attachment

Comments on the Quality of English Language

Minor editing of English language required

Author Response

Thank you for your valuable comments, we have carefully revised them. Please see the attachment.

Reviewer 2 Report

Comments and Suggestions for Authors

This paper proposed a residual useful life prediction method based on the spatial-temporal similarity calculation method to improve the remaining useful life prediction accuracy in case of the complex operating conditions of aviation turbofan engine data set and the original noise of the sensor. Before accepting, the article still has the following issues that need to be clarified.

1. What exactly is 'spatial characteristics on mechanical life prediction' (line 90)? For turbofan engines, what factors cause this characteristic? What adverse effects will it have on turbofan engines and their RUL? What is the influencing mechanism?

2. In Section 2, the article gives many definitions of formulas and variables. I want to know what is your starting point for defining these variables, namely, what problem is each variable introduced to solve?

3. line 255, it should be ‘Figure 4’.

4. Is the RUL estimation method proposed in this paper universal, namely, can it be extended to other industrial processes?

5. The simulation in this paper is a little thin. Figures 9&10 and Table 6&7 show the same simulation results, but one is a histogram and the other is a table, which is meaningless.

6. The parameters (such as the number of network layers, nodes, topological structure, etc.) of the comparison algorithms SVR, and CNN…are not explained, which cannot guarantee that the results obtained are the best ones for these comparison algorithms, reducing the confidence of the simulation.

7. When selecting parameters, the author uses a data analysis method to select some available features (line 250), including core speed. But for turbofan engines, fan speed is also an important state parameter. About 80% of the thrust of a turbofan engine is generated by the bypass, which is closely related to fan speed. In the actual industrial process, whether in condition monitoring or control, fan speed is an important parameter. Is it reasonable in terms of the physical principle not to choose fan speed as the characteristic parameter here?

Comments on the Quality of English Language

Minor editing of English language required

Author Response

(The authors gave the same response as above.)

Reviewer 3 Report

Comments and Suggestions for Authors

An interesting paper has been submitted for consideration. The Authors proposed a novel data-driven method of life prediction of turbofan engines. The paper is well-written, used math seems to be ok. I have only a few remarks:

1) Page 3 "Definition 3..." should start as a new line.

2) Change the writing style in Fig. 1 caption: "model ... model"

The above comments yield the review to recommend the paper for publication after minor corrections.

Comments on the Quality of English Language

The quality of English is good. The paper is easy to read. Minor spell check is required.

Author Response

(The authors gave the same response as above.)
